# What Is Considered a Variation of Biomechanical Parameters in Tensile Tests of Collagen-Rich Human Soft Tissues?—Critical Considerations Using the Human Cranial Dura Mater as a Representative Morpho-Mechanic Model

**DOI:** 10.3390/medicina56100520

**Published:** 2020-10-05

**Authors:** Johann Zwirner, Mario Scholze, Benjamin Ondruschka, Niels Hammer

**Affiliations:** 1Department of Anatomy, University of Otago, Dunedin 9016, New Zealand; 2Institute of Materials Science and Engineering, Chemnitz University of Technology, 09125 Chemnitz, Germany; mario.scholze@mb.tu-chemnitz.de; 3Department of Macroscopic and Clinical Anatomy, Medical University of Graz, 8010 Graz, Austria; 4Institute of Legal Medicine, University Medical Center Hamburg-Eppendorf, 22529 Hamburg, Germany; b.ondruschka@uke.de; 5Department of Orthopaedic and Trauma Surgery, University of Leipzig, 04103 Leipzig, Germany; 6Fraunhofer IWU, 01187 Dresden, Germany

**Keywords:** biomechanical parameters, elastic modulus, mechanical variation, ultimate tensile strength, tensile testing

## Abstract

*Background and Objectives:* Profound knowledge on the load-dependent behavior of human soft tissues is required for the development of suitable replacements as well as for realistic computer simulations. Regarding the former, e.g., the anisotropy of a particular biological tissue has to be represented with site- and direction-dependent particular mechanical values. Contrary to this concept of consistent mechanical properties of a defined soft tissue, mechanical parameters of soft tissues scatter considerably when being determined in tensile tests. In spite of numerous measures taken to standardize the mechanical testing of soft tissues, several setup- and tissue-related factors remain to influence the mechanical parameters of human soft tissues to a yet unknown extent. It is to date unclear if measurement extremes should be considered a variation or whether these data have to be deemed incorrect measurement outliers. This given study aimed to determine mechanical parameters of the human cranial dura mater as a model for human soft tissues using a highly standardized protocol and based on this, critically evaluate the definition for the term mechanical “variation” of human soft tissue. *Materials and Methods:* A total of 124 human dura mater samples with an age range of 3 weeks to 94 years were uniformly retrieved, osmotically adapted and mechanically tested using customized 3D-printed equipment in a quasi-static tensile testing setup. Scanning electron microscopy of 14 samples was conducted to relate the mechanical parameters to morphological features of the dura mater. *Results:* The here obtained mechanical parameters were scattered (elastic modulus = 46.06 MPa, interquartile range = 33.78 MPa; ultimate tensile strength = 5.56 MPa, interquartile range = 4.09 MPa; strain at maximum force = 16.58%, interquartile range = 4.81%). Scanning electron microscopy revealed a multi-layered nature of the dura mater with varying fiber directions between its outer and inner surface. *Conclusions:* It is concluded that mechanical parameters of soft tissues such as human dura mater are highly variable even if a highly standardized testing setup is involved. The tissue structure and composition appeared to be the main contributor to the scatter of the mechanical parameters. In consequence, mechanical variation of soft tissues can be defined as the extremes of a biomechanical parameter due to an uncontrollable change in tissue structure and/or the respective testing setup.

## 1. Introduction

Mechanical parameters obtained from human tissues are fundamental to accurately simulate the load deformation behavior of these tissues in computer simulations [1,2] and physical replicas [3,4]. Further to this, direct comparison of characteristic mechanical properties allows to compare for age [5], sex [6] or site-dependent differences [7] of biological tissues when loaded. The elastic modulus (E_mod_) is a mechanical parameter describing the ratio of stress and strain under small deformation assuming a linear elastic behavior of the respective tissue [8], and consequently the rigidity of a tissue when elastically deformed; it is one of the key parameters to mechanically simulate human tissues in computer models [1,9,10]. Ultimate tensile strength (UTS) characterizes the maximum stress that is applicable to a tissue in relation to its cross-sectional area before it fails when stretched continuously [8]. UTS is suitable for comparisons of different graft materials for transplant purposes considering the mechanical resistance of the tissue [11]. Strain at maximum force (SF_max_) describes the elongation of a tissue at the point of the maximum applicable load in relation to the tissues initial unloaded length; this seems useful to understand the contribution of individual components such as cells to the overall mechanical behavior of the respective tissue [11,12]. Generally, baseline datasets to obtain the aforementioned mechanical parameters in tensile tests reveal widespread variation throughout the body [5,6,7,13]—an accepted “expectable” condition for human tissues. However, several factors such as the clamping quality [14,15], strain rate [16] and the mechanical and bio-physico-chemical tissue structure [5,17,18] impact the mechanical parameters to a varying extent (Figure 1). Consequently, it remains unclear to date what the term “variation” describes in the context of mechanical parameters retrieved from human tissues related to their morphological characteristics.

This given study for the first time aims to define the variation in the context of mechanical parameters of soft tissues, deploying human cranial dura mater (DM) as a representative model. The cranial DM is a collagen-rich tissue and its biomechanical behavior is of interest for transplant surgery [11] or lifelike human head models that are used to study impact scenarios [19]. Controlling a variety of testing procedure and tissue-related factors to the best possible extent, the following hypothesis was investigated: biomechanical parameters of human soft tissues are reproducible, if factors that influence these parameters are kept strictly constant.

## 2. Materials and Methods

### 2.1. Retrieval and Processing of Human Temporal Dura Mater Samples

A total of 124 human temporal DM samples (44 left, 80 right side) of 75 donors (26 ♀, 49 ♂) were retrieved from an avascular area located between the anterior and posterior branches of the middle meningeal artery. Age and post mortem interval (PMI) between death of the individuals and harvesting the samples averaged 50 ± 24 years (range 3 weeks to 94 years) and 71 ± 31 h (range 11 to 146 h), respectively. All samples were retrieved in a fresh and chemically unfixed state during routine forensic autopsies at the Institute of Legal Medicine, University of Leipzig, Germany and stored in a −80° C freezer following an initial precooling step at 4 °C. The retrieval and use of the tissues for the given purpose was approved by the Ethics Committee of the University of Leipzig, Germany (protocol number 486/16-ek).

### 2.2. Adjustment of Water Content

After thawing, the samples were cut into a dog bone shape using a scalpel and a 3D-printed template adapted from the ISO 527-2 standard [19]. The macroscopically visible collagen bundles of the bone surface layer were being orientated along the load application axis in the shaft area. Following this, all samples were placed in sealed dialysis membranes of 64 mm (Spectra/Por^®^; molecular weight cut 6–8 kDa) and submerged into a 5 wt% 20 mM hydroxymethyl aminomethane-buffered polyethylene glycol (Tris-PEG; pH = 7.4; Merck KGaA, Darmstadt, Germany; molecular weight 20,000 Da) solution for 24 h on a shaking table as described previously [5].

### 2.3. Mechanical Testing

Prior to mechanical testing, the samples were molded with polysiloxane impression material (medium-bodied, Exahiflex; GC Corporation, Tokyo, Japan) in the middle of the shaft of the dog bone for the determination of cross-sectional areas and thicknesses. Casts of the cross-sectional areas were scanned on a commercial scanner (Perfection 7V750Pro; Seiko Epson Corporation, Suwa, Japan) at a resolution of 1200 dpi and subsequently calculated with the Measure 2.1d software (DatInf, Tübingen, Germany). Small subsamples of eight samples were taken prior to mechanical testing to determine their individual water content by means of the lyophilization technique [20]. For the standardized and tight mounting of all samples, customized 3D-printed clamps were used [14]. Black graphite powder was applied to the naturally bright surface of the DM samples to qualitatively assess the tissue slippage during the performed mechanical tests. For this purpose, a single charge-coupled camera with a resolution of 2.8 Megapixels (Q400; Limess, Krefeld, Germany) was mounted perpendicular to the surface of the sample. Uniaxial tensile tests were performed using a universal testing machine (Allround Table Top Z020; Zwick Roell, Ulm, Germany) and an Xforce P load cell of 2.5 kN (ISO 7500 accuracy grade 1; accuracy < 1%, repeatability < 1%, reversibility < 1.5%, zero error < 0.1%, resolution < 0.5%—all five criteria are fulfilled from 0.4% of the maximum force, which corresponds to 10 N) with testControl II measurement electronics (all Zwick Roell) at an ambient temperature of 22 °C. All samples were preconditioned with 20 load-unload cycles at a force range of 0.5 to 2.0 N, before loading at a crosshead displacement rate of 20 mm/min was applied until failure. All DM samples were strained in the longitudinal axis according to the predominant collagen orientation that was macroscopically visible on the surface of the sample.

### 2.4. Scanning Electron Microscopy

Scanning electron microscopy (SEM) was conducted on 14 representative samples using a JEOL 6700F field emission scanning electron microscope (JEOL, Peabody, Boston, MA, USA). The samples were treated in a K575X sputter coater with a 5-nm layer of gold-palladium (Emitech Technologies, Kent, England) as a preparation step for the SEM. The bone surface layer and arachnoid layer were scanned of six samples each for a qualitative assessment of the collagenous structure of the respective sample. The cross section was scanned of two samples to qualitatively assess the internal collagenous structure of the DM samples.

### 2.5. Data Processing and Statistical Analyses

The stress-strain curves were calculated from the crosshead displacement data. E_mod_, UTS and SF_max_ were assessed as depicted in Figure 2. For statistical evaluation, Excel Version 16.16 (Microsoft Corporation, Redmond, WA, USA) and GraphPad Prism version 8 (GraphPad Software, San Diego, CA, USA) were used. An Anderson–Darling normality test was used to assess Gaussian distribution of the data. A one-way ANOVA was used to compare parametric data of samples and a Kruskal–Wallis, followed by an uncorrected Dunn’s test for nonparametric data. Age, PMI, sex, thickness and side of the samples were correlated to the biomechanical parameters obtained in this study determining the Pearson or Spearman correlation coefficient according to the Gaussian distribution of the data. The water contents of the eight small samples were correlated to the biomechanical parameters of the respective specimens they were taken from. P values equal to or smaller than 0.05 were considered statistically significant.

## 3. Results

All of the 124 tested samples passed the preconditioning cycles and, therefore, could be included in the statistical evaluation. This was directly related to a proper clamping of the samples and their overall ability to withstand loads of at least 2 N. The clamping quality was deemed “high” as only neglectable specimen slippage occurred, which was indicated by an absence of broad unspeckled areas close to the upper and lower clamps post testing. In addition, all tested samples failed in the shaft area of the dog bone. The medians and interquartile ranges (IQRs) for the cross-sectional areas and thicknesses of the samples were 2.84 mm^2^ (IQR = 1.21 mm^2^) and 0.87 mm (IQR = 0.37 mm).

### 3.1. Elastic Modulus

Values for the E_mod_ were available from 77 samples as provided by the testControl II software. The median E_mod_ was 46.06 MPa (IQR = 33.78 MPa; Figure 3). The minimum and maximum E_mod_ were 10.51 MPa and 116.4 MPa, resulting in a 23% to 253% variation of the median. The E_mod_ correlated negatively with the thickness of the sample (*p* = 0.005; r = 0.315) on a statistically significant level, but was not significantly correlated with age (*p* = 0.226), sex (*p* = 0.743), side (*p* = 0.853), PMI (*p* = 0.180) and water content (*p* = 0.343). None of the samples with the same thickness presented identical E_mod_, but a range of different values (see the red points as examples in Figure 3).

### 3.2. Ultimate Tensile Strength

Values for the UTS were obtained from all 124 samples. The median UTS was 5.56 MPa (IQR = 4.09 MPa; Figure 4). The minimum and maximum UTS were 1.23 MPa and 21.68 MPa, resulting in a 22% to 390% variation of the median. UTS negatively correlated with the age of the donors at a highly significant level (*p* ≤ 0.001; r = 0.285), but was statistically insignificant with regards to sex (*p* = 0.782), side (*p* = 0.794), thickness (*p* = 0.073), PMI (*p* = 0.470) and water content (*p* = 0.169). A 60-year-old male (PMI: 44 h; side: left; thickness = 0.87 mm) and a 60-year-old female (PMI: 62 h; side: left; thickness = 0.84 mm) were the only tested samples of the entire dataset to reveal an identical UTS of 6.64 MPa. Apart from this a range of different values was observed, even for samples with similar anthropometric data (see the green points in Figure 4).

### 3.3. Strain at Maximum Force

The SF_max_ was obtained from 124 samples. The median SF_max_ was 16.58% (IQR = 4.81%; Figure 5). The minimum and maximum SF_max_ were 6.62% and 29.03%, resulting in a 40% to 175% variation of the median. The SF_max_ statistically significantly and negatively correlated with the age of the donors (*p* ≤ 0.001; r = 0.227), but was non-significantly correlated to sex (*p* = 0.822), side (*p* = 0.516), thickness (*p* = 0.247), PMI (*p* = 0.364) and water content (*p* = 0.171). A range of different values was observed (see the purple points in Figure 5) for samples of the same age as representatively shown for 28-year-old samples in Figure 5. Over the entire dataset a 60-year-old male (PMI: 44 h; side: left; thickness = 0.87 mm) and a 60-year-old female (PMI: 62 h; side: left; thickness = 0.84 mm) were the only samples to show an identical SF_max_ of 19.95%.

### 3.4. Scanning Electron Microscopy

SEM revealed that the human DM consists of multiple collagen layers from superficial (bone surface layer) to deep (arachnoid layer; Figure 6). The bone surface layer of human DM showed areas with both isotropic and anisotropic collagen organization (Figure 6).

## 4. Discussion

Load-deformation properties of human soft tissues are typically accompanied by large variations [7,11,13,21,22] with standard deviations or interquartile ranges of the E_mod_ and UTS frequently being larger than half of the mean value depending on the normal distribution of the experimental data [6,11,18,20,23,24,25]. The here tested human cranial dura mater is a thin membranous layer that is softly attached to the inside of the skull bone. It mainly consists of densely packed collagen bundles with some scattered fibroblasts and elastic fibers all being embedded in a mucopolysaccharide-water matrix [26,27,28].

### 4.1. The Experimental Side Involved in the Mechanical Variation of Human Soft Tissues

In the given study, it was assured to keep all factors influencing the load-deformation properties of human soft tissues constant to the best possible extent. For this purpose, a combination of techniques was deployed, allowing for a highly standardized measuring setup and tissue preparation. Regarding the experimental setup, 3D-printed clamps with sharp pyramids qualitatively assured only minimal specimen slippage and mounting in highly standardized specimen dimensions [14]. All of the tested samples in this study passed the preconditioning procedure, with failure occurring in the shaft area of the dog bone shaped samples during testing. Therefore, the clamping quality that is provided by custom made 3D-printed clamps [14] can be deemed “high” for wet human soft tissues with thicknesses ranging between 0.5 and 2.2 mm, as the here used DM. Hence, clamping related bias may only contribute little to the here observed scatter of the mechanical properties. Likewise, the testing velocity as the second contributing factor to the measuring setup is considered to be insignificant as a contributor to the scatter of mechanical parameters, considering the viscoelastic and time dependent nature of biological tissues. This is assumed as the here applied quasi-static testing velocity of 20 mm/min was comparatively low and very likely to be delivered reliably by the machine for all tested samples. The here given setup calculated the mechanical parameters based on the crosshead displacement data using a 3D-printed template that allowed to taper all the samples to the same size, which is specified above. However, a digital imaging correlation (DIC)-based approach [5] should be chosen, when it is intended to use the given mechanical parameters for, e.g., computer modelling purposes rather than for the here given intention. Using DIC, the mechanical parameters can be measured based on in-field strains of the stochastic speckle pattern on the surface of the sample. An important advantage of DIC is that even small differences in the inter-individual shape of the samples can be balanced when the mechanical parameters are calculated, which is not possible with the here given approach. Therefore, DIC-based data provides more accurate strain values compared to crosshead displacement-based data. If the sample does not present a natural stochastic pattern, which applies for most biological materials, this pattern has to be created on the surface of the sample, with potential effects on the mechanical parameters (e.g., increase of elastic modulus by dehydrating effect of sample coating or denaturation effect on collagens with resulting strength decrease). For the human iliotibial tract as another collagen-rich soft tissue, surface coating with water- and solvent-based coating sprays as well as graphite speckling was shown not to affect the mechanical parameters of the same [29].

### 4.2. The Influence of Chemical Treatment onto the Variation of Mechanical Parameters

A chemical treatment of the soft tissues such as the use of fixatives for cadaveric preservation may alter the collagen interaction and hence the mechanical parameters of the respective tissue, which was shown for the human scalp, DM and temporal muscle fascia [18,30,31], or iliotibial tract specimens when fixed with ethanol or formaldehyde [24]. Formaldehyde as a component of the Thiel fixation is known to irreversibly block the amino groups of peptides and to cross-link hydroxymethylene bridges [32,33].

### 4.3. Water Content Does Influence the Mechanics of Human Soft Tissues

Beyond these measurement setup-related parameters, the biomechanical properties of soft tissues in tensile tests are influenced by the mechanical [5,11,12] and bio-physico-chemical [17] tissue composite of the samples. Recently, it was shown in human iliotibial tract samples that higher water contents were associated with both lower values for E_mod_ and UTS [17]. However, the SF_max_ remained largely uninfluenced by the water content. For this given study, an average water content of the samples was determined following osmotic adaptation combined with a pH-value-based buffering using Tris-PEG. Despite this effective measure to osmotically adjust the water content, it is important to underline that both in vivo and post-mortem there is unlikely a particular value, rather than a range of values. This might be caused by the here used osmotic stress protocol including measuring errors occurring during the lyophilization technique [20] or by the varying potential of the tissues to bind water. To assure that all samples were constantly submerged and to keep the fluid moving without the build-up of sediments, all tissues were placed on a shaking table and exposed to a gentle moving while being submerged in the fluid. Additionally, the water content of the here tested samples is influenced by drying following their removal from the Tris-PEG for the testing procedure itself. Environmental conditions such as temperature and humidity affect the hydration state of the tested sample. In this given study, temperature and hydration state potentially affected the reported E_mod_ and UTS values to an unforeseeable extent within the time frame between removal of the sample from the Tris-PEG bath and reaching the maximum force of the sample during the tensile test. The maximum force is the peak force value that is used to calculate the here reported mechanical parameters. Environmental chambers that allow to control temperature and humidity [34,35] during the mechanical tests provide the best possible control of these factors. Albeit the processing time has been kept standardized and minimal for the samples, given the different specimen thicknesses and the total amount of water, drying could have influenced the measurements to varying extent. A standardized handling of the samples was further assured by the highly standardized 3D-printed testing equipment and clamping procedures, which do not require time-consuming manual closure. Moreover, the preconditioning assured to align the extracellular matrix perpendicular to the direction of load application, which is known to increase the reproducibility of the obtained mechanical parameters by alleviating post mortem-, storage-, freezing-, thawing- or fixation-related influences [36].

While a measurement bias due to the handling and subsequent measuring of the tissues to determine the respective water content appears to be manageable, the capacity of the tissues to bind water likely varies remarkably even within the same tissue type due to their morphological features. The here used DM is a multi-layered composite tissue that mainly consists of compact and dense water-attracting collagens and few elastic fibers, which are embedded in glycosaminoglycans and water [5,28]. Cellular components such as fibroblasts, vessels or mesothelial cells are scattered in the compact extracellular scaffold of the human DM [27,37]. It remains unclear to date how the individual DM substructures contribute to the overall hydration of the tissue and, thus, the tested sample. Therefore, even though in the given study the tissues were removed from vessel-free DM areas in a uniform manner, it remains challenging to control the biological factors that contribute to the water content of each tested sample and to standardize the mechanical testing accordingly. Based on biomechanical testing of acellular matrices it was concluded that collagens are the main load-bearing elements of matrix-rich soft tissues with cells having a negligible influence on the overall biomechanical behavior of the tissue [11,12].

### 4.4. Extracellular Matrix as a Composite Structure Biases the Mechanical Parameters of Soft Tissues

The collagen architecture of soft tissues such as the DM is largely determined by their highly individual mechanical loading and consequently continuous remodeling throughout life [27]. This renders it unlikely to retrieve soft tissues with an identical number of load-resistant elements from different individuals and additionally align the same in an exactly reproducible manner for biomechanical tests based on macroscopic criteria. Therefore, extracellular matrix interactions may have also contributed to the scatter of the mechanical values in the given study, even though the samples were handled uniformly. The decrease in SF_max_ with increasing age, which was noted for the DM samples in this study can be interpreted as an increase in collagen interaction or collagen-proteoglycan interactions with age that progressively limit their respective straining potential, which was, however, only insignificantly related to the E_mod_ and UTS in this study.

Further to the described biochemical nature of the tissue composite, the mechanical properties of soft tissues are influenced by mechanical characteristics of the tissue structure, namely the amount and arrangement of load-resistant elements. This is especially important when separated areas and small volumes are tested mechanically. More precisely, the size of a tested sample impacts the results and strength if it does not contain a representative volume with the same composition as the remaining tissues. Small volumes could furthermore contain higher amounts of collagens or extracellular matrix compared to other samples, which introduces errors already when harvesting the samples. The influence of the collagen amount on the biomechanical parameters of soft tissues was previously demonstrated by comparing the almost entirely collagenous temporal muscle fascia to the temporal muscle with a collagen content of only about 1% [38,39]. The temporal muscle was significantly more elastic and less resistant to the applied load [39]. However, as the particular collagen content of each tested sample was not tested, its influence on the scatter of the biomechanical parameters of samples of the same tissue type remains unclear. Freeze-thaw cycles of the samples between retrieval from the cadaver and mechanical testing might destruct the collagens or alter the collagen interaction, with potential implications on the biomechanical properties of the tested samples. Depending on the thawing environment and the potentially associated evaporation of water, an osmotic adaption after freezing is recommended to ensure a similar hydration state of all tested samples, as was done here. However, the effect of freeze-thaw cycles on the here reported biomechanical properties of human soft tissues is unclear to date as only few studies are existing on biological tissues in this matter in general with only very limited sample sizes [40,41]. In this study, no repeated thawing cycles were necessary until testing to reduce the influence of this temperature change to the best possible extent. Lastly, the commonly anisotropic collagen fiber orientation within a tested sample considerably influences its biomechanical properties. It was shown that collagen fibers resist higher loads when stretched aligned rather than perpendicular or orientated in a random fashion [42]. Soft tissues such as the here investigated DM are commonly multi-layered and the orientation of deeper layers in relation to the axis of load application cannot be reliably deduced based on the macroscopically visible orientation of the most superficial layer. Therefore, the overall orientation of collagens within a tested sample remains uncontrollable and, thus, the collagen arrangement very likely contributes to the scatter of the biomechanical parameters. In this regard, to ensure the highest possible comparability between different studies of the commonly anisotropic biological tissues, the testing of the respective samples should be reported with respect to the superficially visible predominant fiber orientation. If the fiber orientation is respected in the related studies, soft tissue samples are predominantly strained along the predominant fiber course that is macroscopically visible on the surface of the sample [39,43]. However, even when the superficial fibers are used as an orientation of the underlying predominant fiber orientation, the resulting orientation of the tested sample with respect to its overall predominant fiber orientation remains approximative and most likely largely subjective.

### 4.5. Biological Variation Biases the Mechanical Variation to Larger Extent than the Experimental Setup

Considering all factors which influence the mechanical parameters of human soft tissues in tensile testing and the limited predictability of the mechanical and chemical structure of a tissue, it is highly unlikely that the overall test conditions remain exactly reproducible and hence identical biomechanical properties are obtained. This is exemplified in this study by an UTS value obtained from the left side of one donor, which was almost five times as large compared to its contralateral counterpart. Determining of the impact of anatomical variation on the biomechanical parameters remains challenging, as no identical numerical value can be obtained for soft tissues with contemporary methods, which complicates an intra- and inter-individual comparison. The correlation of quantitative morphological data such as the mean collagen orientation throughout the entire tested sample with the obtained biomechanical properties might provide further insights into the morpho-mechanics of human collagen-rich soft tissues. While the destructive nature of SEM investigations is unfeasible in this regard, confocal microscopy that uses the collagenous autofluorescence [44] might be a promising option for future studies. Altogether, we have to reject our hypothesis and state that the mechanical parameters of soft tissues remain impossible to be exactly reproduced due to a variety of uncontrollable factors of mainly the tissue structure. The variation seems to be parameter-specific and can be expected to be in the range of 23% to 253% (E_mod_), 22% to 390% (UTS) and 40% to 175% (SF_max_) around the median according to the results of the given study. Based on the findings of this study the following definition is established for the mechanical variation of human soft tissue parameters: “An uncontrollable bias based on differences in their structural composition and alignment and/or a bias caused by the used experimental setup”.

### 4.6. Limitations

The sample size of this study was limited, which had been restricted by the available number of tissues for the given project. The samples were retrieved from the temporal DM as a model tissue for the given purpose. The mechanical behavior of other soft tissues might differ and impact the conclusions that can be drawn. Further to this, the impact of systemic diseases on the biomechanical parameters of the here investigated tissues remains unknown. The here presented E_mod_ and SF_max_ values are based on the crosshead displacement data rather than in-plane surface strain measurements, which are considered more accurate and, therefore, should be preferred when using these parameters for computational modeling purposes [5]. Here, the E_mod_ was used to compare the load-deformation behavior of the different DM samples. However, human soft tissues often present a non-linear, e.g., a hyper-elastic behavior. Therefore, the E_mod_ likely represents an oversimplification of the true load-deformation behavior of the here presented DM samples. The determination of cross-sectional areas was performed by molding with polysiloxane impression material in the middle of the shaft area between the upper and lower clamping region according to previous publications [5,14]. It has to be noted that due to, e.g., air bubbles during the setting time of the material the determined cross-sectional area might end up being too large compared to the real one, which might have directly influenced the E_mod_ and UTS stated in this study. The performance of the mechanical tests in an environmental test chamber might have reduced the scatter of the data. However, the time between the removal of the tested sample from the Tris-PEG solution and the end of the tensile test was in the range of two to three minutes. The influence of deep-freezing for storage purposes on the mechanical parameters in these studies is unknown. Even though a similar composition of tissue samples of the same origin can be expected, these tissues most likely at least slightly differ in their composition such as the cell-extracellular matrix ratio and their individual thickness, which should be considered when the mechanical properties of different samples are compared. The given work critically considered different factors that influence any biomechanical experiment of this kind. Even though it was intended to give the readers a good overview, this work does not claim to be an encompassing literature review on the given topic. The here given study used crosshead displacement data to evaluate the mechanical parameters of the human dura mater in order to demonstrate the variation of biomechanical parameters in tensile tests of collagen-rich tissues. However, for computational modelling purposes, DIC-based data as provided in other studies [5] should be used as the resulting absolute mechanical values can be considered more accurate and appropriate for this purpose.

## 5. Conclusions

Biomechanical properties of soft tissues are highly variable in spite of all attempts to minimize the bias introduced by the experimental setup. Tissue morphology appears to be the main contributor to the scatter in the load-deformation properties of collagen-rich soft tissues in quasi-static tensile testing. In consequence, the variation of biomechanical properties of collagen-rich soft tissues forms an omnipresent phenomenon in biomechanics and has to be considered when biomechanical parameters of these tissues are obtained.

## Figures and Tables

**Figure 1 medicina-56-00520-f001:**
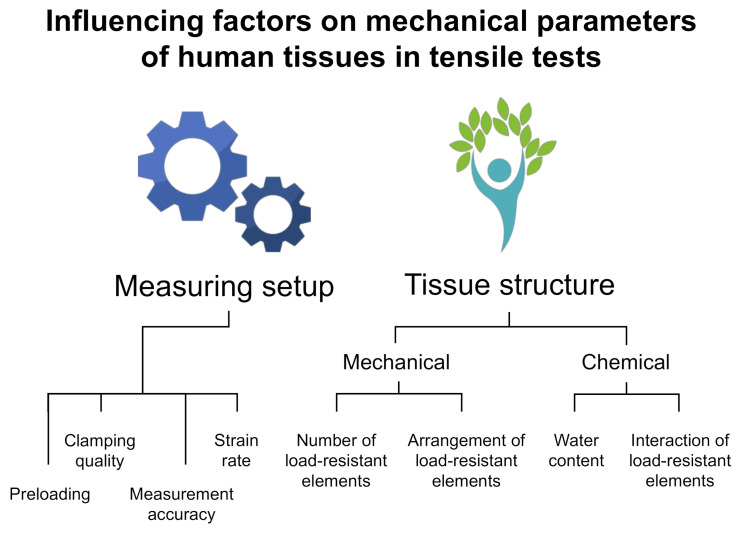
Major influencing factors on the mechanical parameters of human tissues in tensile tests are depicted.

**Figure 2 medicina-56-00520-f002:**
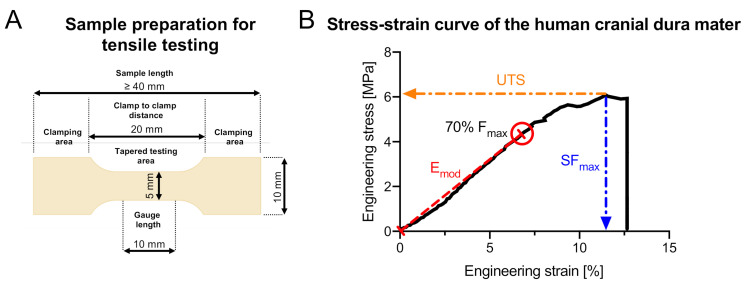
(**A**) The dimensions of the dog bone-shaped cranial dura mater specimens are depicted. (**B**) The determination of the biomechanical parameters in this study is shown in a representative stress-strain curve of one of the cranial dura mater samples that were tested in this study. The elastic modulus (E_mod_) was calculated through a linear regression analysis between the zero-point and the point that equals 70% of the maximum force (F_max_). Ultimate tensile strength (UTS) is the maximum stress (F_max_ divided by the cross-sectional area) before tissue failure when being strained. Strain at maximum force (SF_max_) depicts how much the sample was strained at the point of the UTS compared to its initial length.

**Figure 3 medicina-56-00520-f003:**
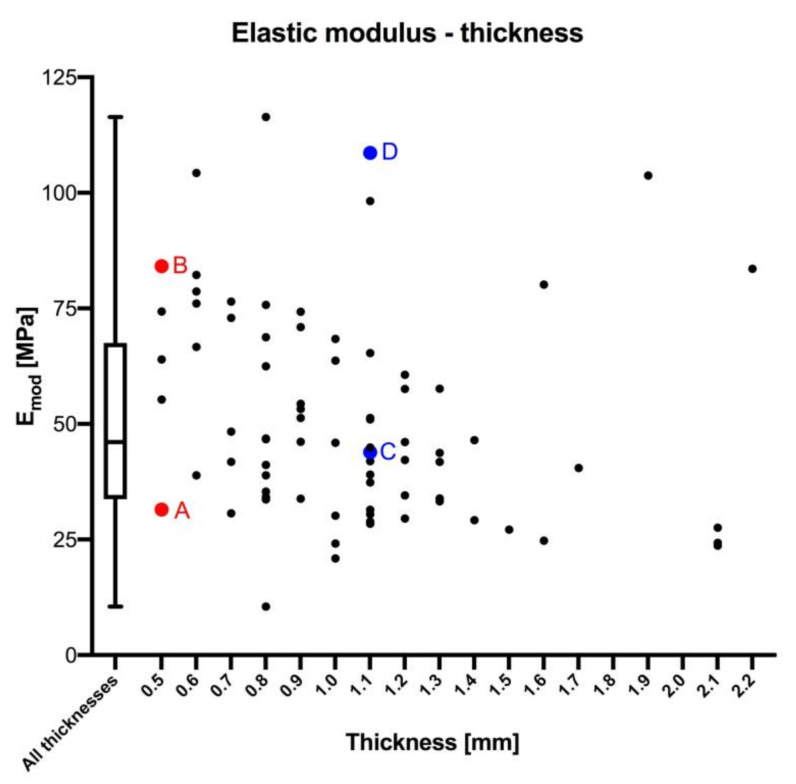
The elastic modulus is depicted in relation to the thickness of dura mater samples. Additional data to the colored sample pairs (**A**–**D**, pairs of an identical thickness) are: (**A**) age: 50 years, post mortem interval (PMI): 104 h, elastic modulus (E_mod_) = 31.50 MPa, sex: male, side: left, water content: 80.93%; (**B**) age: 36 years, E_mod_ = 84.15 MPa, PMI: 88 h, sex: female, side: right, water content: 70.77%; (**C**) age: 77 years, E_mod_ = 43.87 MPa, PMI: 57 h, sex: male, side: right, water content: 74.72%; (**D**) age: 43 years, E_mod_ = 108.62 MPa, PMI: 120 h, sex: female, side: right, water content: 81.17%.

**Figure 4 medicina-56-00520-f004:**
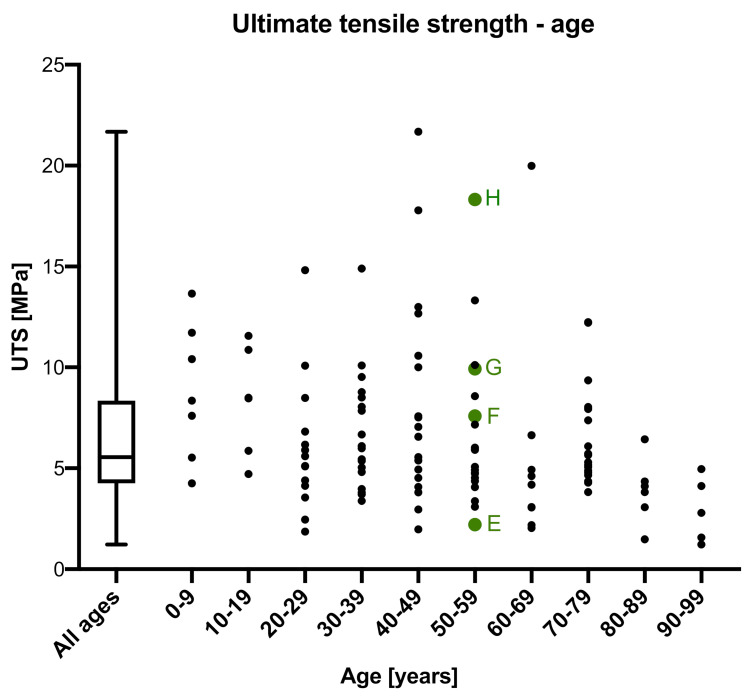
The ultimate tensile strength (UTS) is depicted in relation to the age of the dura mater samples. Additional data to four age-matched samples of two 50-year-old donors (**E**–**H**) are: (**E**) post mortem interval (PMI): 104 h, sex: male, side: right, thickness = 0.7 mm, UTS = 2.21 MPa,; (**F**) PMI: 122 h, sex: female, side: right, thickness = 0.9 mm, UTS = 7.56 MPa; (**G**) PMI: 104 h, sex: male, side: left, thickness = 0.5 mm, UTS = 9.93 MPa; (**H**) PMI: 122 h, sex: female, side: left, thickness = 0.8 mm, UTS = 18.32 MPa.

**Figure 5 medicina-56-00520-f005:**
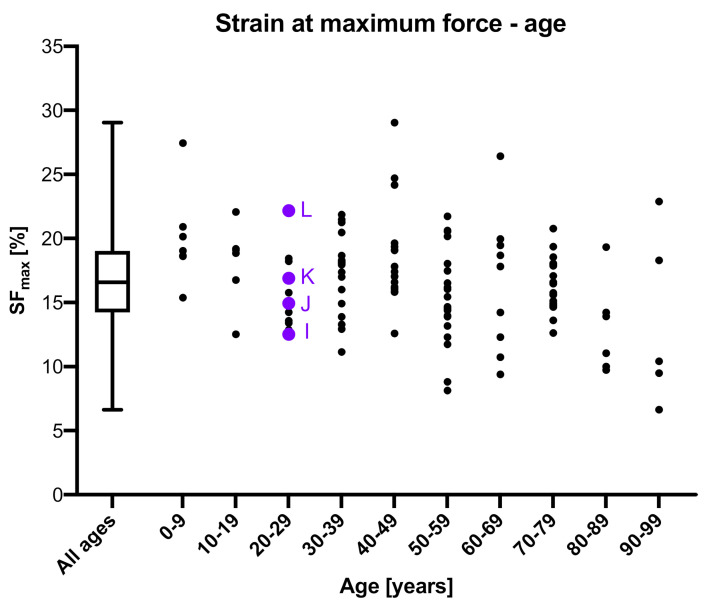
Strain at maximum force (SF_max_) is depicted in relation to the age of the dura mater samples. Additional data to the four 28-year-old purple colored samples (**I**–**L**) are: (**I**) post mortem interval (PMI): 63 h, sex: male, side: right, SF_max_ = 12.52%, thickness = 2.1 mm; (**J**) PMI: 34 h, sex: male, side: right, SF_max_ = 14.93%, thickness = 0.6 mm; (**K**) PMI: 70 h, sex: male, side: left, SF_max_ = 16.88%, thickness = 0.8 mm; (**L**) PMI: 14 h, sex: male, side: left, SF_max_ = 22.16%, thickness = 1.1 mm.

**Figure 6 medicina-56-00520-f006:**
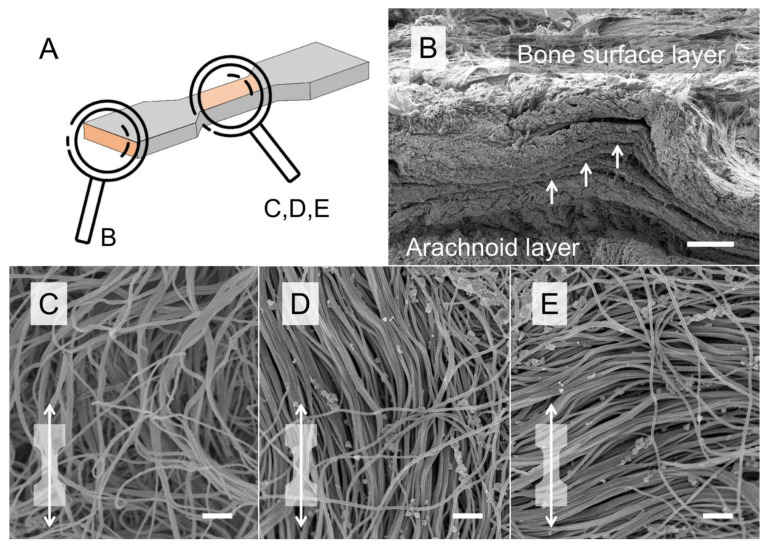
Representative scanning electron microscopy (SEM) images of the human temporal dura mater (DM) are depicted. (**A**) Scanning orientation of samples B–E, (**B**) a cross-sectional SEM scan is depicted showing the multi-layered organization of the human DM. Scale bar, 100 μm. (**C**–**E**) SEM scans of the DM bone surface layer reveal isotropic (**C**) and anisotropic (**D**,**E**) areas. The dog bone shape with the white arrow indicates that samples are strained along the axis (**D**) or perpendicular to the axis (**E**) of aligned collagens within the shaft area of the dog bone-shaped sample during tensile tests. Scale bars, 1 μm.

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
