# Peer review of "What Is Considered a Variation of Biomechanical Parameters in Tensile Tests of Collagen-Rich Human Soft Tissues?—Critical Considerations Using the Human Cranial Dura Mater as a Representative Morpho-Mechanic Model"

_medicina, 2020, doi:10.3390/medicina56100520_

Round 1
Reviewer 1 Report
What is considered a variation of biomechanical parameters in tensile tests of human soft tissues? – Critical considerations using the human dura mater as a morpho-mechanic model
Authors
Zwirner , Scholze , Ondruschka , Hammer
Lines 41-56: As a literature review, the anatomy seemed vague and a bit rushed. Assuming that scholars will be reviewing this content, it is to the point. The gross anatomy and relevance to surgery and nerve block may be beneficial to medicine and science.
Profound knowledge on the load-dependent behavior of human soft tissues is required for the development of suitable replacements as well as for realistic computer simulations.
Lines 61-69: Provide a good identification to article selection and the exhaustive research performed. Specific key words and categorizations are identified.
Regarding the former, e.g. the anisotropy of a particular biological tissue has to be represented with site- and direction-dependent particular mechanical values.
Lines 70-82: Case series are to the point. The process is clear and consise, with an emphasis on research components.
Parameters of soft tissues scatter considerably when being determined in tensile tests.
Lines 87-91: The review of literature, identifies exactly what the authors are searching for, connecting to a synthesis of literature.
Setup- and tissue-related factors remain to influence the mechanical parameters of human soft tissues to a yet unknown extent.
- “It is to date unclear if measurement extremes should be considered a variation or whether these data have to be deemed incorrect measurement outliers.”
- Conclusions: “It is concluded that mechanical parameters of soft tissues such as human dura mater are highly variable even if a highly standardized testing setup is involved.”
Lines 92- 113:
The authors identify the aim of the study. Upper and lower subscapular nerve information is identified with connection between the literature. Barriers are addressed in this section,
As explained in the abstract, “This given study aimed to determine mechanical parameters of the human dura mater as a model for human soft tissues using a highly standardized protocol and based on this, critically evaluate the definition for the term mechanical “variation” of human soft tissue.”
Materials and Methods: A total of 124 human dura mater samples with an age range of 3 weeks to 94 years were uniformly retrieved, osmotically adapted and mechanically tested using customized 3D-printed equipment in a quasi-static tensile testing setup.
The illustrations are concise and identify the focus of research. Case series from lines 143-162 explain the content in greater detail, with imaging to confirm colorings and 3D printed component.
Results: The here-obtained mechanical parameters were scattered (elastic modulus = 46.06 MPa, interquartile range = 33.78 MPa; ultimate tensile strength = 5.56 MPa, interquartile range = 4.09 MPa; strain at maximum force = 16.58%, interquartile range = 4.81%).
Lines 163-186: Discussion presents findings that are well explained scientifically and potential benefits to science and research.
As stated, “The tissue structure and composition appeared to be the main contributor to the scatter of the mechanical parameters.” Lines 187- 191 address bias and limitations for this research.
“In consequence, mechanical variation of soft tissues can be defined as the extremes of a biomechanical parameter due to an uncontrollable change in tissue structure and/or the respective testing setup”. The authors explain future study focuses and investigations, lines 198-200.
Reviewer 2 Report
The paper entitled « What is considered a variation of biomechanical parameters in tensile tests of human soft tissue ? - Critical considerations using the human dura mater as a morpho-mechanic modeling » deals with the influence of test conditions, tissue architecture and composition on the mechanical properties of dura mater samples.
The paper consists of an introduction with a non-exhaustive bibliography on soft tissue mechanical testing in general, a material and method section describing tissue sampling, adjustement of water content, mechanical testing, SEM observations and data analysis. A results section highlights the influence of age, side, sex, PMI and water content on elastic modulus (Emod), ultimate stress and strain at failure and some SEM observations. A discussion section comments on the dispersion of the values whose origin could be due to the experimental conditions, (grip quality in the jaws etc), the chemical treatment of the samples, the water content and orientation of the collagen fibers and biological variations of the samples. The content of the paper focuses on different influences on the mechanical behavior of dura mater tissues but does not really propose any modeling based on these observations. The title referring to a mechanical morpho-mechanical model seems more ambitious than the actual content of the paper.
The characterization of biological tissues is confronted with the difficulty of experimental dispersion, which can be related to the variety of inter- and intra-individual properties, but also to the experimental conditions to the composition of the tissues. It is often very tricky to separate these different influences and clearly identify the impact of one of these elements in relation to another because the tests cannot free themselves from defects in experimental set-up, sample orientation, inhomogeneity of composition or orientations, etc. The subject of this paper is very interesting, but it presents some inaccuracies that do not allow us to verify the fundamental hypotheses.
Paragraph line 51 « The elastic modulus (Emod) is a mechanical parameter describing the ratio of stress and strain [8] and consequently the resistance of a tissue when elastically deformed. »
Elastic modulus is a mechanical parameter describing the ration between stress and strain under small deformation (it is the tangent modulus at origin) with hypothesis of linear elastic behaviour. This article aims at studiying the mechanical behaviour of soft tissues. Soft tissues have often non linear behaviour and representing their behavious only with Elastic modulus might be reductive. Moreover the Elastic modulus physically represents the rigididty of the material, the resistance is more described by the ultimate stress. In my opinion, authors should be more rigourous on the definition of the mechanical parameters and their choice to reduce their study to Elastic Modulus and linear elastic behaviour which is not an evidence when studying soft tissue.
Paragph line 60 to 65 : The environemental condition also have an impact on identified mechanical properties. Tests performed on dry sample or in water as well as temperature influence (lab temperature, 37°C) should also impact especially when tests take a long time. The storage conditions should also impact the mechanical properties one might have different results on fresh or frozen samples.
The size of « load resistant element» regarding to the size of the tested sample should also impact results if Elementary Representative Volum is not obtained.
The precision of the orientation of the sample especially for anisotropic tissue is often approximative, with approximative visual orientation before cutting might also induced dispersion.
The homogeneity of the sample should also impact the dispersion, is the thickness constant ? are the fibers homogenously distributed etc. ? Authors have done SEM observations where mean orientations of fibers as well as dispersion in orientations are easilly quantificable. Why not having use this information and correlated to mechanical properties ? It is a pity that this information, shown as being of primary importance in the first figure, has not been exploited.
Line 88 How the sample are cut in a dog bone shape ? I imagine the specimen are cut out with a punch ? Dimensions of the specimens should be given, maybe just add dimensions to the plan of figure 5A.
A stress-Strain curve should be ploted in order to justify the choice of linear elastic mechanical parameter. The choice to use a linea elastic model in small deformation to characterize soft tissue without ploting a curve to justify the mechanical behaviour and validity of this assumption on with all results of the paper is funded is an important gap that does not allow the reader to qualify the validity of the results presented.
In addition to the non-justification of the choice of the mechanical model, the identification methods were not detailed. The method to define Emod should be explained : soft material have often non linear mechanical behaviour. One can compare behaviour at small deformation with the Emod at orgine but the range of data used for identification has an important impact on result and the method for defining the linear part should be detailed (often a R² criteria on linearity is used for defining the data identification range).
Line 109 The class and precision of the load cell should be given as well as the maximum force reached during the tests.
Line 111 Could you explain the objective of preload cycles ?
Line 138 The cross-sectional areas and thickness are very small, is there a Representative Elementary Volume ?
Biological samples often show geometric and structural inhomogeneities (variations in thickness within the same sample), fiber bundle sizes or orientation dispersion. All this can lead to inhomogeneities in the deformation fields that can make the calculation of the deformation from the displacement of the crosshead unusable and require the use of full field measurements or localized measruement method in interest area of the samples. The authors have all the information to answer these questions (observation of the orientations of collagen fibers in the samples, variation in thickness from the scan in each specimens, tests filmed by a camera, etc) but none of this information is used quantitatively, even though it can question the initial hypotheses and influence the results.
Figure 2 a small error appears in the abscissa scale bar 2.1 and 2.0 should be reversed.
Line 235 « be » should be replaced by « by »
Round 2
Reviewer 2 Report
The authors have fully integrated the remarks and justified their answers with which I generally agree.
In their response, the authors indicated that they measured the deformations by digital image correlation. This point does not appear in the experimental protocol of mechanical testing. I think that the authors should specify this point (method of measurement, software used). This allows the reader to have a complete and accurate view of the experimental protocol and allows other researchers to be able to repeat similar tests in another study on other tissues using a similar protocol. Thus it is important to be exhaustive in the description of experimental protocols to maximize the impact of publications.
Except this minor precision I accept the manuscript for publication.
